# Simultaneous Production of Cellulose Nitrates and Bacterial Cellulose from Lignocellulose of Energy Crop

**DOI:** 10.3390/polym16010042

**Published:** 2023-12-21

**Authors:** Ekaterina I. Kashcheyeva, Anna A. Korchagina, Yulia A. Gismatulina, Evgenia K. Gladysheva, Vera V. Budaeva, Gennady V. Sakovich

**Affiliations:** Bioconversion Laboratory, Institute for Problems of Chemical and Energetic Technologies, Siberian Branch of the Russian Academy of Sciences (IPCET SB RAS), 659322 Biysk, Russia; yakusheva89_21.ru@mail.ru (A.A.K.); julja.gismatulina@rambler.ru (Y.A.G.); budaeva@ipcet.ru (V.V.B.);

**Keywords:** *Miscanthus × giganteus*, cellulose, enzymatic hydrolysis, cellulose nitrates, bacterial cellulose

## Abstract

This study is focused on exploring the feasibility of simultaneously producing the two products, cellulose nitrates (CNs) and bacterial cellulose (BC), from *Miscanthus × giganteus*. The starting cellulose for them was isolated by successive treatments of the feedstock with HNO_3_ and NaOH solutions. The cellulose was subjected to enzymatic hydrolysis for 2, 8, and 24 h. The cellulose samples after the hydrolysis were distinct in structure from the starting sample (degree of polymerization (DP) 1770, degree of crystallinity (DC) 64%) and between each other (DP 1510–1760, DC 72–75%). The nitration showed that these samples and the starting cellulose could successfully be nitrated to furnish acetone-soluble CNs. Extending the hydrolysis time from 2 h to 24 h led to an enhanced yield of CNs from 116 to 131%, with the nitrogen content and the viscosity of the CN samples increasing from 11.35 to 11.83% and from 94 to 119 mPa·s, respectively. The SEM analysis demonstrated that CNs retained the fiber shape. The IR spectroscopy confirmed that the synthesized material was specifically CNs, as evidenced by the characteristic frequencies of 1657–1659, 1277, 832–833, 747, and 688–690 cm^−1^. Nutrient media derived from the hydrolyzates obtained in 8 h and 24 h were of good quality for the synthesis of BC, with yields of 11.1% and 9.6%, respectively. The BC samples had a reticulate structure made of interlaced microfibrils with 65 and 81 nm widths and DPs of 2100 and 2300, respectively. It is for the first time that such an approach for the simultaneous production of CNs and BC has been employed.

## 1. Introduction

Cellulose nitrates (CNs), being the initial product from the chemical functionalization of cellulose, have been the subject of active study for nearly 200 years [1]. This is associated with the global abundance of natural cellulose and humankind’s high need for mold plastics, quality lacquers, printing inks, biomolecular adhesive membranes, and energetic binder. The global market keeps growing, especially due to the demand for CNs as the platform for biosensors in analytical medicine [2]. CNs have acquired particular importance in disease diagnostics and treatment due to their microporous structure and strong affinity for interacting with and subsequently absorbing a biomaterial (for example, antibodies) [2,3,4]; antiCOVID-19 masks [5] and composite filter membranes for oligonucleotide extraction [6,7] have emerged. The energetic properties of CNs are in demand as constituents of explosive compositions [8,9,10,11] in the mining industry, road construction in mountainous areas, and focused demolition of obsolete structures because the safety and handling issues associated with CN-based compositions have currently been resolved at a very high level [12]. It should be emphasized that CNs themselves have become precursors of more complex chemicals with unique energetic characteristics [13]. An overview of the published data on the demand for CNs in the industry allows for the conclusion that there is an increasing need for CNs with a nitrogen content ranging from 10.6% to 12.0%.

The cellulose nitration process itself, as an example of the chemical functionalization of the most naturally abundant, easily renewable biopolymer, is of particular importance for fundamental science [14]. Cellulose is a naturally made polymer of β-glucose whose units are bound by a 1-4-β-glycosidic linkage and is produced in the amount of about 1.3 × 10^9^ tons a year via photosynthesis [15]. In the cellulose nitration process, the hydrogen atom of the (–H–OH) hydroxyl is replaced by the (–NO_2_) nitro group when cellulose is nitrated by nitrating mixtures. It is possible to synthesize CNs with a wide range of functional properties by varying the nitrating mixture composition and nitration process parameters [1,15,16].

Due to the deficiency of conventional cotton and wood raw resources [14], research on the possibility of producing new CN types with tunable functional properties starting from alternative raw sources is currently growing more urgent. The list of non-woody cellulose sources of vegetal origin continues to expand actively. The studies conducted in some countries have demonstrated the possibility of synthesizing CNs with satisfactory properties from cellulose isolated from Alfa grass [17], Rhizophora, kenaf, palm oil bunches (EFB) [16,18], giant reed [9,19], tobacco (Nicotiana tobacum) stalks [20], giant panda feces [21], Posidonia oceanica Brown Algae [22], hemp [23], intermediate flax straw [24], oat hulls [25], and *Miscanthus sinensis* (Andersson) var. Soranovskii [26].

Many researchers compare the properties of alternative cellulose with the requirements for cotton cellulose: the α-cellulose content must be no less than 92%. The above-listed studies on nitration differ in their focus: some researchers only reflect nitration process conditions and specify one of the properties of the resultant nitrocellulose, basically the CN degree of substitution [10,20], while others report extensive information including basic functional, supramolecular, morphological, and energetic characteristics of the biopolymers synthesized from a specific feedstock type [9,19,21,22].

The possibility of producing a new CN type from bacterial cellulose (BC) has been justified alongside the use of non-woody cellulose, and prospects for using this new type have been considered [10,27,28,29]. 

BC has a chemical structure similar to plant-based cellulose but does not contain hemicelluloses, lignin, or pectin. BC microfibrils form a three-dimensional network, ensuring high values of mechanical strength, degree of polymerization, crystallinity, and water-holding capacity [30,31]. The structure and those properties allow BC to have numerous potential technical applications: environmentally friendly electronic devices [31], flexible organic light-emitting diodes, fuel cells, flexible supercapacitors, headphones, monitors, and materials for electromagnetic wave absorption [32].

The studies on BC as a precursor of CNs are associated with excellent and unique nanoscale properties [30,33,34,35,36], with hydrolyzates from pretreated lignocellulose being examined as the nutrient medium [36,37,38,39,40,41,42,43,44]. There are examples of hydrolyzates derived from *Miscanthus* [32,45]. Additionally, there is a need to upscale the biosynthesis process to ensure sufficient amounts of the precursor for promising CNs [46], but studies of this kind are scarce. 

The overview of the world literature on alternative cellulose sources has demonstrated that despite the diversity of *Miscanthus* species and their habitats [47,48,49,50,51,52,53], as well as their wide applications ranging from paper industry to biosynthesis products [54,55,56,57,58,59,60,61], the information on *Miscanthus* cellulose as the substrate for chemical functionalization into CN is presently quite limited, except for the authors’ single studies.

All the methods for cellulose production from non-woody sources, including *Miscanthus*, pursue the aim of conforming to the purity of cotton cellulose: the α-cellulose content no less than 92%, the pentosan content no more than 2%, and minimal lignin; therefore, there is no information on enzymatic hydrolysis as the pretreatment method for a cellulosic product to modify its structural characteristics with subsequent nitration. Even more so, there are no examples of utilizing cellulose to directly produce CNs and concurrently as the source of a glucose nutrient medium for subsequent biosynthesis. 

The present study aimed to explore whether the two standalone products, CNs (a cellulose chemical modification product) and BC (an enzymatic hydrolysis product), could concurrently be produced from the energy crop *Miscanthus × giganteus*. 

## 2. Materials and Methods

All the reagents and materials used in this study were procured from AO Vekton (Saint-Petersburg, Russia).

### 2.1. Feedstock

In this study, *Miscanthus × giganteus* was used as the feedstock, having the following chemical compositions: 50.2 wt.% Kürschner cellulose, 19.5 wt.% acid-soluble lignin, 21.2 wt.% pentosans, 1.63 wt.% ash, and 0.5 wt.% extractives [53]. The quantitative determination methods for the feedstock components were similar to those used for cellulose components (Section 2.2.1), except for the quantitative determination of Kürschner cellulose and extractives. Kürschner cellulose was determined by the extraction of the *Miscanthus* sample with 1:4 mixed nitric acid–alcohol [62,63]. Extractives were determined by extracting the sample in dichloromethane using a Soxhlet extractor according to the Technical Association of the Pulp and Paper Industry (TAPPI) standard [64].

### 2.2. Preparation and Analysis of Cellulose Samples 

Since the grinding size plays a decisive role in cellulose isolation [65,66], *Miscanthus × giganteus* was ground on a KR-02 fodder grinder (TechnoMash, Miass City, Russia) to a particle size of 2–12 mm prior to use. 

Cellulose was isolated by the nitric-acid method involving treatment of the weighed portion of the feedstock with dilute solutions of nitric acid (3–6%) and sodium hydroxide (3–6%) under atmospheric pressure at 90–95 °C for 6–8 h and 4–6 h, respectively [26].

#### 2.2.1. Analysis of Chemical Composition and Cellulose Degree of Polymerization (DP) 

The chemical composition (contents of α-cellulose, lignin, ash, and pentosans) and cellulose DP were analyzed by standard chemical and physicochemical methods. The α-cellulose content of the cellulose sample was determined as per the TAPPI standard by treating cellulose with a 17.5 wt.% NaOH solution, followed by the quantification of the undissolved residue after cellulose was washed with a 9.5 wt.% NaOH solution and water and dried [67]. Klason lignin (acid-insoluble lignin) was measured as per TAPPI T222 om-83 [68]. Pentosans were quantified by transforming the same in a boiling 13 wt.% HCl solution into furfural, which was collected in the distillate and determined on a xylose-calibrated UNICO UV-2804 spectrophotometer (United Products & Instruments, Dayton, NJ, USA) calibrated against xylose (a 630-nm wavelength) using orcinol-ferric chloride [69]. The ash content was quantified by cellulose incineration in accordance with TAPPI T211 om-85 [70]. The cellulose DP was determined from the outflow time of cellulose solution in cadoxene (cadmium oxide in ethylenediamine) in a VPZh-3 viscometer (OOO Ecroskhim, Moscow, Russia) with a capillary diameter of 0.92 mm [71].

#### 2.2.2. X-ray Diffraction Analysis of Cellulose Samples

X-ray examination of the cellulose sample was performed on a DRON-6 monochromatic diffractometer (Burevestnik company, Nalchik City, Russia) with Fe-Kα radiation at 3 to 145° scattering angles in reflection and transmission geometries at room temperature [43,72,73].

The degree of crystallinity (DC) was defined as the relation between the integrated scattering intensity from the crystalline phase and the total integrated scattering intensity from the crystalline and amorphous phases in reflection geometry (Equation (1)): (1)$$DC=\frac{I_{c}-I_{am}}{I_{C}}\times 100\;\%$$
where *I_c_* is the total integrated scattering intensity from the crystalline and amorphous components and *I_am_* is the integrated scattering intensity from the amorphous component [43,72].

### 2.3. Enzymatic Hydrolysis of Cellulose Samples

Enzymatic hydrolysis of the cellulose sample was performed with an enzyme cocktail of Ultraflo Core (Novozymes A/S, Bagsværd, Denmark) and CelloLux-A (Sibbiopharm Ltd., Berdsk, Russia) at a dosage as follows: Ultraflo Core 46 FPU/g solid and CelloLux-A 40 FPU/g solid. The cellulase activity expressed in FPU was determined by the reported procedure [74].

The enzymatic hydrolysis was carried out in a 0.05 M acetate buffer (pH 4.7): a 45.0 g/L initial solid loading on a dry matter basis, a 0.1 L acetate buffer volume, 46 ± 2 °C temperature, and a 150-rpm stirring rate. The first stage of hydrolysis included the measurement of the cellulose DP during the process. For this, the enzymatic hydrolysis was run in a parallel manner in seven 0.5-L conical flasks, with a process time of 2, 4, 6, 8, 24, 32, and 48 h. The stirring was carried out using an ECROS PE-6410 horizontal heated stirrer (Ecohim, Moscow, Russia). Once the time elapsed, the flask was removed from the stirring device, and the reaction mixture was cooled and filtered. The concentration of reducing sugars (RS) in the hydrolyzate was measured on a Cary 60 UV-Vis (Agilent Technologies, Santa Clara, USA) at a 530-nm wavelength using 3,5-dinitrosalicylic acid (Panreac, Spain) as the reagent [69,75]. The RS yield was estimated by Equation (2) [74]: (2)$$\eta_{RS}=\frac{C_{RS}}{C_{S}}\times 0.9\times 100$$
where
*ηRS* is the yield of RS on a substrate weight basis (%).*C_RS_* is the final concentration of RS in the hydrolyzate (g/L).*C_S_* is the substrate concentration on a dry matter basis (g/L).0.9 is the factor associated with the water molecule addition to anhydroglucose residues of the respective monomeric units as a result of hydrolysis.


The solid residue after filtration was thoroughly washed, dried, and weighed to calculate the weight loss. The sample was then analyzed for the cellulose DP (Section 2.2.1) and DC (Section 2.2.2).

The second hydrolysis stage involved working up cellulose samples for nitration and preparing nutrient media (hydrolyzates) for BC synthesis. The process was performed in conical flasks under the same conditions. The initial solid loading was 45 g/L, with the reaction mass volume increasing. The weight of the substrate to be hydrolyzed was calculated with allowance for weight loss at the corresponding hydrolysis time and minimal cellulose weight for nitration, and was 3 g. Upon the process’s completion, the resultant reaction mass was filtered, and the solid residue was washed and dried. The solid residue was further analyzed for cellulose and then nitrated. The liquid phase (hydrolyzate) was used for BC synthesis.

To achieve accurate results, three samples were enzymatically hydrolyzed at a time in each experiment. 

### 2.4. Cellulose Sample Analysis after Enzymatic Hydrolysis

The DP, DC, and morphology of the cellulose samples after enzymatic hydrolysis were determined by the same methods as those used for the initial *Miscanthus* cellulose sample (Section 2.2). 

### 2.5. Nitration and Analysis of CN Samples 

The cellulose nitrate samples were obtained by the common sulfuric-nitric acid process using a commercial sulfuric-nitric acid mixture. The cellulose samples were nitrated as follows: The initial water content in the mixture was 14 wt.%, the nitration temperature was 25–30 °C, the nitration time was 40 min, and the mass ratio of substrate to mixed acid was 1:40. The nitration was performed in a 500-mL porcelain beaker with continuous stirring using a HS-50A-Set vertical stirring device (Witeg, South Korea). The nitration temperature was maintained using a water bath.

After the nitration was completed, the resultant CN samples were separated from the spent mixed acid by using a Büchner funnel and a vacuum pump, then the remaining reaction mixture was expelled with a dilute 25 wt.% mixed acid and further thoroughly washed and subjected to three-step high-temperature stabilization with continuous stirring as follows: boiling in water at 80–90 °C for 1 h, boiling in a 0.03% sodium carbonate solution at 80–90 °C for 3 h, and boiling in water at 80–90 °C for 1 h. After the stabilization process was completed, the target products were washed with distilled water until neutral wash waters and then dried for 24 h in open air at room temperature and then at 100 ± 5 °C for 1 h in a BINDER ED23 drying oven (BINDER GmbH, Tuttlingen, Germany) and analyzed.

The CNs were analyzed using common procedures. The nitrogen content was quantified by the ferrous sulfate method [76,77,78] that involves the saponification of CN with concentrated sulfuric acid and the reduction of the resultant nitric acid with iron (II) sulfate to nitrogen (II) oxide, in which nitric acid in excess of nitrogen (II) oxide produces a complex compound, Fe(NO)]SO_4_, that colors the solution into a yellowish-pink color. The solubility of CN (1 g) in acetone (50 mL) was determined by filtration of the acetone-insoluble CN residue, followed by drying and weighing on an Explorer Pro EP214C analytical balance (Ohaus, Langacher, Switzerland). The viscosity of the CN samples was determined from the outflow time of a 2% acetone solution from a VPZh-3 capillary glass viscometer. The solubility of the CN samples in mixed alcohol/ester was determined by filtration of the CN residue insoluble in mixed alcohol/ester, followed by drying and weighing on an Explorer Pro EP214C analytical balance. 

The yield of the CN samples was calculated by Equation (3): (3)$$W=\frac{\left(m_{pr}\times 100\right)}{m_{init}}$$
where *m_pr_* is the weight of the synthesized CN samples, g; and *m_init_* is the weight of the initial cellulose sample for nitration, g.

### 2.6. Structural Analysis of Cellulose, CN, and Coupled TGA/DTA 

The fiber surface morphology of the cellulose and CN samples was examined by scanning electron microscopy (SEM) on a GSM-840 electron microscope (Jeol, Tokyo, Japan) after sputter-coating a Pt layer of 1−5 nm thick. 

The molecular structure of the cellulose and CN samples was examined by Fourier-transform spectroscopy on an Infralum FT-801 spectrometer (OOO NPF Lumex-Sibir, Novosibirsk, Russia) operating at 4000−500 cm^−1^. To acquire spectra, the samples were pressed into pellets with potassium bromide in a CN:KBr ratio of 1:150.

The thermal behavior of the cellulose and CN samples was examined by thermogravimetric (TGA) and differential thermogravimetric (DTG) analyses using a TGA/DTG-60 thermal analyzer (Shimadzu, Nakagyo-ku, Japan) as follows: a weighed portion of 0.5 g, a heating rate of 10 °C/min, a maximal temperature of 350 °C, and nitrogen as inert medium. 

### 2.7. Synthesis of Bacterial Cellulose 

Symbiotic *Medusomyces gisevii Sa-12,* acquired from the Russian National Collection of Industrial Microorganisms, was used as the microbial producer. The vital activity of *Medusomyces gisevii Sa-12* was maintained in a Binder-400 climate chamber (Berlin, Germany) under static conditions at 27 °C for 7 days in a synthetic glucose medium composed of glucose and black tea extractives [36,79]. The seed material was inoculated as 10 vol.% of the nutrient medium volume, which is equivalent to the following cell count: the total yeast count of at least 12.9–13.2 × 10^6^ cells per 1 cm^3^ and the total acetobacteria count of at least 1.6–2.2 × 10^6^ per 1 cm^3^. 

The biosynthesis of BC was conducted on the enzymatic hydrolyzate under static culture conditions at a temperature of 27 °C, with an initial glucose concentration of 20 g/L and a black tea extract content of 1.6 g/L. The cultivation was carried out in a climate chamber (Binder, Germany) for 10 days.

After the cultivation was completed, the BC gel–film was removed from the surface of the nutrient medium and washed to remove the nutrient medium components and cells through a stepwise treatment with 2 wt.% NaOH and 0.25 wt.% HCl, followed by washing with distilled water until neutral wash waters. The obtained BC films were freeze-dried in an HR7000-M freeze dryer (Harvest Right LLC, Salt Lake City, UT, USA) to constant weight.

The yield of the dried BC was calculated by Equation (4):(4)$$W=\frac{m}{C\times V\times 0.9}\times 100\%$$
where
*W* is the BC yield, %.*m* is the weight of the BC sample on an oven-dry basis, g.*C* is the RS concentration in the medium on a glucose basis, g/L.*V* is the volume of the medium, L.0.9 is the conversion factor due to the water molecule detachment upon the polymerization of glucose into cellulose.


The yeast and acetobacteria cell counts, as well as the concentration of reducing sugars in the nutrient medium after removing the BC film, were measured as described in [38].

As a control, BC synthesis was conducted on a synthetic nutrient medium with a glucose concentration of 20 g/L and an extractive content of 1.6 g/L under similar conditions.

The morphology of the BC samples was investigated using a scanning electron microscope (JSM-840, Tokyo, Japan) equipped with a Link-860 series II X-ray microanalyzer. The microfibril width was calculated using the ImageJ 1.53k software. The DP of BC was determined according to the procedure described in Section 2.2.1.

## 3. Results and Discussion

### 3.1. Properties of Cellulose Sample

In the nitric acid method for cellulose production, the preliminary hydrolysis stage involves breaking the bonds between the main components of the lignocellulosic matrix and partially removing hemicelluloses. The subsequent treatment with a diluted nitric acid solution allows for the almost complete removal of hemicelluloses (2.5%), partial dissolution, oxidation, and nitration of lignin, leading to the formation of nitrolignin. Further alkaline treatment solubilizes the nitrolignin and removes it from the product (0.5%). This cellulose production method allows a high-quality product to be isolated with a cellulose content of 95.6%, a pentosan content of 2%, and total non-hydrolyzables (ash and lignin) of only 0.6%. The resultant sample exhibited a high DP of 1770 and a DC of 64%.

The quality indicators of nitric acid-treated cellulose from *Miscanthus × giganteus* align with those of cellulose samples from other *Miscanthus* species obtained using the same method, except for their lower DP (880–1050) compared to the latter [26].

The absence of data on the quality indicators of cellulose extracted from *Miscanthus × giganteus* using nitric acid or other methods that would allow for the production of high-quality cellulose is due to *Miscanthus × giganteus* being currently in active use for alternative purposes where high-quality cellulose is not required, such as chemical modification. For instance, *Miscanthus × giganteus* is used in the paper industry, where the requirements for cellulose are considerably lower, with the α-cellulose content not exceeding 86% [54,56].

At the same time, there is significant ongoing research into the nitration of alternative cellulosic raw materials. The quality indicators of cellulose derived from *Miscanthus × giganteus* closely approach or even surpass those of cellulose samples obtained from acacia pulp [80], Rhizophora, oil palm bunches, and kenaf fibers [18], which have undergone successful nitration.

### 3.2. Enzymatic Hydrolysis of Cellulose

The first stage of the enzymatic hydrolysis involved investigating the change in the cellulose DP and DC during a 48 h enzymatic hydrolysis. Additionally, the RS concentration increment and the weight loss were evaluated (Table 1). Conducting the hydrolysis for more than 48 h was not reasonable due to the significant weight loss (over 66%) and, consequently, the small weight of the cellulose residue after the hydrolysis, which residue would further be used for nitration.

Throughout the enzymatic hydrolysis process of *Miscanthus* cellulose samples, there was a gradual increase in the weight loss, indicating the hydrolysis of the substrate and a reduction in the weight of the solid residue required for subsequent nitration. In the initial 2 h of hydrolysis, the cellulose DP decreased from 1770 to 1490, while the DC increased from 64 to 72%. The changes in the cellulose properties may be attributed to the random cleavage of β-1,4-glucosidic bonds by endoglucanase, occurring in the less organized regions of cellulose, leading to a decrease in DP and an increase in crystallinity [81,82,83].

In the subsequent hours of hydrolysis, the DP started to increase and reached its maximum value after 48 h of hydrolysis. As a result of the experiment, the cellulose DP did not almost change: 1790 after hydrolysis vs. 1770 for the initial cellulose. The DC increased by 12% after a 48 h hydrolysis. According to [59,84], significant structural changes are not typical of the enzymatic hydrolysis of cellulosic materials. It was emphasized that there was no substantial decrease in DP during the hydrolysis process, whereas DC may slightly increase. It was explained by the fact that the cellulase complex attacks the cellulose chains and hydrolyzes each chain to the end. As a result, neither the DP nor the ratio of the crystalline to the amorphous material changes significantly. 

During the enzymatic hydrolysis process, the concentration of reducing sugars (RS) reached 30 g/L, corresponding to a 60% yield of RS. That said, reducing sugars were generated at 43% of the maximal yield as early as the initial 2 h of the process, after which the hydrolysis rate was slowing down.

The obtained results (Table 1) were used to determine the hydrolysis length in the second stage of the experiment for obtaining cellulose samples for nitration and nutrient media for BC biosynthesis. At this stage, the substrate weight for hydrolysis was calculated based on the weight loss values for the specific hydrolysis duration and the cellulose weight required for nitration (minimum 3 g).

The hydrolysis of the *Miscanthus* cellulose samples was conducted for 2, 8, and 24 h. The time point at 2 h was chosen because the maximal reduction in the DP occurred within that time, despite the RS concentration in the hydrolyzate not reaching the required value for BC synthesis (20 g/L). The 8 h and 24 h time points were chosen because the structural characteristics of cellulose underwent changes, with the RS concentration being above 20 g/L in the hydrolyzates. The results of the second stage of the experiment are presented in Table 2.

As a result of the enzymatic hydrolysis, three cellulose samples (C2, C8, and C24) were obtained, ranging in mass from 3.0 to 3.4 g. These samples differed in their characteristics from the initial *Miscanthus* cellulose sample (DP 1770, DC 64%) and differed from each other (DP ranging from 1510 to 1760, DC ranging from 72% to 75%) and were of interest for the subsequent nitration. Due to the lack of information on similar experiments with *Miscanthus* cellulose, it is challenging to compare the observed changes in DP and DC of the cellulose residues after enzymatic hydrolysis. During the enzymatic hydrolysis, enzymatic hydrolyzates (hydrolysates C2, C8, and C24) differing in RS concentration (ranging from 13.4 to 27.5 g/L) were also obtained and investigated as a nutrient medium for BC biosynthesis. A brief diagram of the experiment is given in the Appendix A.

### 3.3. Nitration

Given the high requirements for cellulose used in the chemical conversion (minimal contents of lignin, hemicellulose, ash content, and other side inclusions [14], the results (Section 3.1) obtained regarding the compositional analysis of cellulose from *Miscanthus* × *giganteus* (C) do not exclude the possibility of its successful chemical modification into cellulose nitrate (CN) with satisfactory functional properties. Table 3 presents the key functional properties of the CN samples obtained from *Miscanthus* cellulose before and after enzymatic hydrolysis.

It also follows from Table 3 that an increase in the duration of enzymatic hydrolysis from 2 h to 24 h resulted in a rise in the nitrogen content of the CN samples from 11.35% to 11.83%, an increase in the viscosity from 94 mPa·s to 119 mPa·s, and an elevation in the yield from 116% to 131%. The increase in the nitrogen content and, consequently, the CN yield might be due to the enhanced reactivity of cellulose as a result of the multiple fragmentation of its units by the enzymes. It should also be emphasized that the viscosity of CN was changing consistently with a change in the DP of the initial cellulose samples, depending on the enzymatic hydrolysis length. Furthermore, it is important to note that regardless of the hydrolysis duration, all the synthesized samples were CN esters, as they had a 100% solubility in acetone [14].

It can be concluded from the obtained data listed in Table 3 that the CN sample synthesized from cellulose and subjected to 24 h hydrolysis exhibited satisfactory functional properties: a nitrogen content of 11.83%, a viscosity of 119 mPa·s [14], as well as a high degree of homogeneity, as its solubility was 94% in an alcohol-ester mixture compared to the other CN samples after enzymatic hydrolysis and compared to the CN sample derived from the initial cellulose.

The obtained results have no global analogues since there is no information available on the production of CN based on cellulose after enzymatic hydrolysis. However, CN derived from *Miscanthus* cellulose after 24 h enzymatic hydrolysis showed similar characteristics to CN derived from another *Miscanthus* species (11.85% nitrogen content and 97% solubility) [28], except for a significantly lower viscosity (18 mPa·s), which is attributed to the initially lower DP of the cellulose (1020).

In addition, the characteristics of the synthesized CN samples in this study align with those of the CN derived from cellulose from acacia pulp [85], rhizophora, palm oil bunches, and kenaf fibers [18], tobacco stems [20], and oat hulls [25].

Figure 1 shows microphotographs (×200 and ×5000 zoom) of *Miscanthus* cellulose samples before and after enzymatic hydrolysis, as well as the CNs synthesized based on them.

The scanning electron microscopy (SEM) analysis showed that the cellulose sample (Figure 1a), extracted from *Miscanthus* × *giganteus* and not subjected to enzymatic hydrolysis, consists mainly of heterogeneous cellulose fibers with varying shapes and sizes, resembling tubes. Besides, the overall mixture contains individual flattened, wide fibers. The surface of the cellulose fibers exhibits micro-roughness. With an increase in the duration of enzymatic hydrolysis from 2 h to 24 h (Figure 1c,e,g), the microphotographs reveal that the cellulose fibers become shorter and the edges become jagged. With a higher magnification (Appendix A), irregular-shaped pores appear on the fiber surface, and the number of pores on cellulose fibers increases.

After treating the cellulose samples from *Miscanthus* × *giganteus* with a sulfuric-nitric acid mixture, the CN fibers primarily retained the shape of the original cellulose fibers while increasing in volume. The surface of the cellulose-nitrate fibers became smoother. The CN fibers of the sample derived from the original *Miscanthus* × *giganteus* cellulose (Figure 1b) represented separate tube-like fibers, in contrast to the CN samples based on cellulose after the 2–24 h enzymatic hydrolysis (Figure 1d,f,h), which consisted of a mixture of fibers varying in size and shape.

Figure 2 presents the Fourier-transform infrared spectroscopy results for the cellulose and CN samples.

According to Figure 2a, the FTIR spectra of the original cellulose samples exhibit the main functional groups characteristic of cellulose [9,86], namely 3341–3363 cm^−1^, 2898–1901 cm^−1^, 1428–1430 cm^−1^, 1158–1163 cm^−1^, and 1059–1060 cm^−1^, which are assigned to the O–H stretching, asymmetric and symmetric stretching of C–H, O–H bending of absorbed water, asymmetric bending vibration of CH_2_, C–O–C stretching, skeletal stretch of C–O, and vibration of the β-glycosidic linkage of cellulose, respectively^.^ The FTIR spectra showed that the cellulose samples did not exhibit peaks corresponding to the stretch vibrations responsible for impurity components like aromatic structures of lignin at around 1500 cm^−1^ and hemicelluloses at around 1700 cm^−1^, proving once again that the cellulose extracted from *Miscanthus* × *giganteus* was high-quality.

The FTIR spectra of the CN samples (Figure 2b) exhibit the main functional groups that indicate the formation of low-substituted nitrocellulose ethers (1657–1659 cm^−1^, 1277 cm^−1^, 832–833 cm^−1^, 747 cm^−1^, 688–690 cm^−1^). The intense absorption bands in the range of 1657–1659 cm^−1^ correspond to the vibrations of ν_a_(NO_2_) nitrate groups, which are associated with the CH_2_ groups of the glucopyranose rings in the CN (position C(6)). The intense absorption bands at 1277 cm^−1^ can be attributed to the stretching symmetric vibrations of nitrate groups. The absorption bands in the ranges of 832–833 cm^−1^, 747 cm^−1^, and 688–690 cm^−1^ correspond to the vibrations of nitrate groups: stretching ν_a_(NO_2_), wagging γ_w_(NO_2_), and scissoring δ(NO_2_) vibrations, respectively. In addition to the main absorption bands associated with the stretching vibrations of nitrate groups, there are peaks of stretching vibrations of ν(OH) in the range of 3200–3700 cm^−1^, appearing as a broad, complex contour. This indicates the incomplete substitution of the CN. The peaks of stretch vibrations in this region belong to the hydroxyl groups of the CN, which participate in hydrogen bonding and are a characteristic feature of the chemical heterogeneity of the ester. Identical functional groups are observed in the FTIR spectra of CN derived from other alternative plant-based raw materials [9,22,25,87].

Figure 3 shows the TGA/DTG thermograms of the original cellulose samples and their CNs.

According to Figure 3a, the obtained TGA curves for cellulose samples before and after the enazymati hydrolysis can be divided into three distinct regions. The first region encompasses the temperature range from the beginning of the experiment to 100 °C, during which the samples undergo drying, exhibiting a weight loss of 0.2–0.8% accompanied by an endothermic peak. The second region extends from 100 °C to 400 °C, where the samples undergo decomposition with a weight loss of 88.6–90.7% and an associated endothermic transformation. The third region spans the range from 400 °C to 450 °C, where the samples continue to decompose with a minor weight loss of 1.4–1.7%. The temperature range for the onset of intensive sample decomposition was determined to be 339–345 °C.

From the analysis of the literature data, it is well known that higher initial decomposition temperatures correspond to higher thermal stability and purity of the original cellulose [14]. The DTA curves of the cellulose samples (Figure 3c) showed that the decomposition endothermic peak corresponds to a temperature range from 357 °C to 371 °C, with a weight loss of the samples up to 88.6–90.7%, confirming their purity. These results are consistent with the findings of a study on cellulose derived from bitter bamboo stems [87], which showed the superior thermal stability of cellulose from Giant *Miscanthus*.

In the case of the CNs obtained from the cellulose samples before and after enzymatic hydrolysis, as determined by TGA (Figure 3b), it was found that regardless of the hydrolysis duration, the decomposition peak of the CN samples occurred at a temperature around 198–199 °C, and the decomposition continued up to a temperature of approximately 260 °C, with a weight loss of the samples ranging from 70.1 to 82.8%. Further decomposition of the samples occurred with a minor weight loss in the range of 6.9–9.5%.

The DTG curves obtained (Figure 3d) illustrate a single narrow exothermic peak at a temperature around 198–199 °C. Comparing the DTG curves of the CN samples with the curves of the original cellulose samples, it is evident that the temperatures of the exothermic peak in the CN samples decrease from 357–371 °C to 198–199 °C. This destructive behavior is associated with the thermolytic cleavage of the weakest O-NO_2_ group, initiating autocatalytic decomposition and leading to the formation of reactive radicals that accelerate the thermal decomposition process of the nitrated polymer chains [9]. The above findings indicate that the obtained CN samples are chemically pure, high-energy biopolymers. Comparing the obtained TGA/DTA data for the CN synthesized from Giant *Miscanthus* cellulose with those for the CNs derived from cotton [87,88], giant reed [9,19], brown algae [22], and bitter bamboo stems [21], indicates their close correspondence. Furthermore, it is demonstrated that all CN samples exhibit high specific decomposition heats ranging from 6.53–8.28 kJ/g.

Thus, the CN samples obtained from cellulose subjected to enzymatic hydrolysis are low-substituted nitric esters of cellulose with satisfactory functional properties and energetic characteristics. Overall, the synthesized CN samples exhibit properties that indicate the suitability of cellulose samples after enzymatic hydrolysis for chemical functionalization into complex cellulose ethers. It is important to emphasize that this approach to obtaining CN is being used for the first time in global practice.

### 3.4. Synthesis of Bacterial Cellulose 

Biosynthesis of BC was conducted on enzymatic hydrolyzates obtained after 2, 8, and 24 h. Enzymatic hydrolyzates C8 and C24 with RS concentrations of 22.8 g/L and 27.5 g/L, respectively, were adjusted to a concentration of 20 g/L through dilution. Enzymatic hydrolyzate C2 with a RS concentration of 13.4 g/L was also used for BC biosynthesis. The results of enzymatic hydrolysis are presented in Figure 4.

It can be observed from the presented data that the count of yeast at the end of the biosynthesis process exceeds that of acetobacteria in all cases. This can be attributed to the fact that the utilized producer is a consortium of various yeast and acetobacteria species and genera. According to literature data, yeast synthesizes ethanol to stimulate the growth of acetobacteria, which, in turn, produce BC to protect the yeast from the surrounding environment [89,90]. Figure 4 indicates that in the synthetic nutrient medium (control) and in the C8 and C24 hydrolyzates, the count of acetobacteria remains relatively constant, ranging from 8–10 million CFU/mL. A low count of acetobacteria, 1 million CFU/mL, is observed in the nutrient medium of hydrolyzate C2. The low count can be attributed to the RS concentration of 13.4 g/L, which is insufficient for the active growth and vitality of acetobacteria. The low count of acetobacteria resulted in the absence of BC biosynthesis, which, in turn, explains the lack of BC gel film in the nutrient medium of hydrolyzate C2.

Figure 4b shows the residual RS concentration in the culture medium after 10 days of cultivation. The RS concentration after 10 days of cultivation in the synthetic nutrient media (control) was less than 4 g/L, while in the nutrient media of the enzymatic hydrolyzates, it ranged from 8 to 10 g/L. The slight decrease in RS concentration in the nutrient medium of hydrolyzate C2 during the biosynthesis process, from 13.4 g/L to 10 g/L, indicates the absence of active vitality in the acetobacteria responsible for BC production. The high residual RS concentration in nutrient medium C2 compared to C24 is explained by the low concentration of acetobacteria (Figure 4a) and, as a consequence, by the low consumption of RS in nutrient medium C2, indicating the absence of the active viability of acetobacteria.

The BC yield in the nutrient media of hydrolyzates C8 and C24 was 11.1% and 9.6%, respectively. This obtained yield is high and comparable to the control yield of 11.8%. These results indicate the preservation of BC yield when transitioning from a synthetic medium to nutrient media derived from the cellulose hydrolyzates of *Miscanthus*. A BC yield of 10% is not considered low. For example, when using Kombucha Original Bio as a producer, the BC yield in the synthetic nutrient medium (control) and in apple waste nutrient medium was 1% and 4%, respectively [44], which is 10 to 11 times lower than the BC yield obtained in our study on the hydrolyzates. BC yields ranging from 9.6% to 11.1% highlight the advantage of Medusomyces gisevii Sa-12 over individual strains that can yield BCs at only 2.2–6.5% [36,91,92,93].

The morphology of the BC samples synthesized on synthetic nutrient medium (control) and enzymatic hydrolyzates was investigated by SEM (Figure 5). The overall morphological structure of the BC samples exhibited an intertwined network of microfibrils with inter-fibrillar spaces, consistent with the structure of BC samples reported in the literature [46,94,95]. The width of the microfibrils for the BC sample synthesized on the synthetic nutrient medium (control) ranged from 26.0 nm to 229.0 nm, with an average width of 58.0 nm. The width of the microfibrils for the BC samples synthesized on enzymatic hydrolyzates C8 and C24 ranged from 24.0 nm to 186.0 nm, with an average width of 65.0 nm for C8 and 81.0 nm for C24, indicating values close to the control. The width of microfibrils in BC samples can depend on the nature of the producer or the composition of the nutrient medium [44,96]. Therefore, in our case, the nutrient medium composition does not have a significant influence on this characteristic.

The DP of the BC samples synthesized on enzymatic hydrolyzates was determined to be 2100 for C8 and 2300 for C24, compared to 2500 for the control. These values are relatively high and similar to each other [97,98].

Thus, it has been established that enzymatic hydrolyzates C8 and C24 are suitable for obtaining high-quality BC samples.

## 4. Conclusions

Research has been conducted on the possibility of simultaneous production of two independent products from *Miscanthus × giganteus* cellulose: CNs and bacterial cellulose. Precursors for CNs and nutrient media for bacterial cellulose (BC) synthesis were obtained through an incomplete enzymatic hydrolysis of the *Miscanthus* cellulose sample for 2, 8, and 24 h. The solid residues obtained after hydrolysis, which were cellulose samples, differed in their structural characteristics from each other (DP 1510–1760, DC 72–75%), as well as from the original cellulose sample (degree of polymerization of 1770 and crystallinity of 64%). Nitration of the cellulose samples revealed that all precursors were suitable for chemical functionalization, as evidenced by the complete solubility (100%) of the synthesized CN in acetone. Prolonging the duration of enzymatic hydrolysis from 2 to 24 h resulted in a subsequent 0.48% increase in the nitrogen content of CN and a 15% yield increase. It was found that the maximum duration of enzymatic hydrolysis (24 h) led to the production of CN samples with satisfactory functional properties: nitrogen content of 11.83%, viscosity of 119 mPa·s, and solubility in a mixed alcohol/ester and diethyl ether mixture of 94%. SEM showed that during the nitration process, the fibers of the CN samples became smoother, retained the shape of the original cellulose fibers, and exhibited a slight increase in volume. FTIR spectroscopy demonstrated that the obtained CN were low-substituted nitrate esters of cellulose, as all spectra contained major functional group frequencies associated with nitro groups at 1657–1659 cm^−1^ and 1277 cm^−1^.

The enzymatic hydrolysis of *Miscanthus* cellulose samples for 2, 8, and 24 h resulted in hydrolyzates with reducing sugar concentrations ranging from 13 to 28 g/L. It was found that nutrient media based on the hydrolyzates obtained after 8 and 24 h were of good quality and provided high BC yields of 11.1% and 9.6%, respectively. Scanning electron microscopy (SEM) revealed that the obtained BC samples had a mesh-like structure composed of nanoscale fibrils. The average width of the microfibrils in the BC samples synthesized using the 8 h hydrolyzate was 65.0 nm, while it was 81.0 nm for the 24 h hydrolyzate, which was close to the synthetic nutrient medium (control) at 58.0 nm. The DP of the BC samples was relatively high, measuring 2100 and 2300, respectively, which was slightly lower than the control at 2500. This approach of simultaneous production of CNs and BC has been applied for the first time and tested on lignocellulose from an energy plant, yielding unprecedented results.

## Figures and Tables

**Figure 1 polymers-16-00042-f001:**
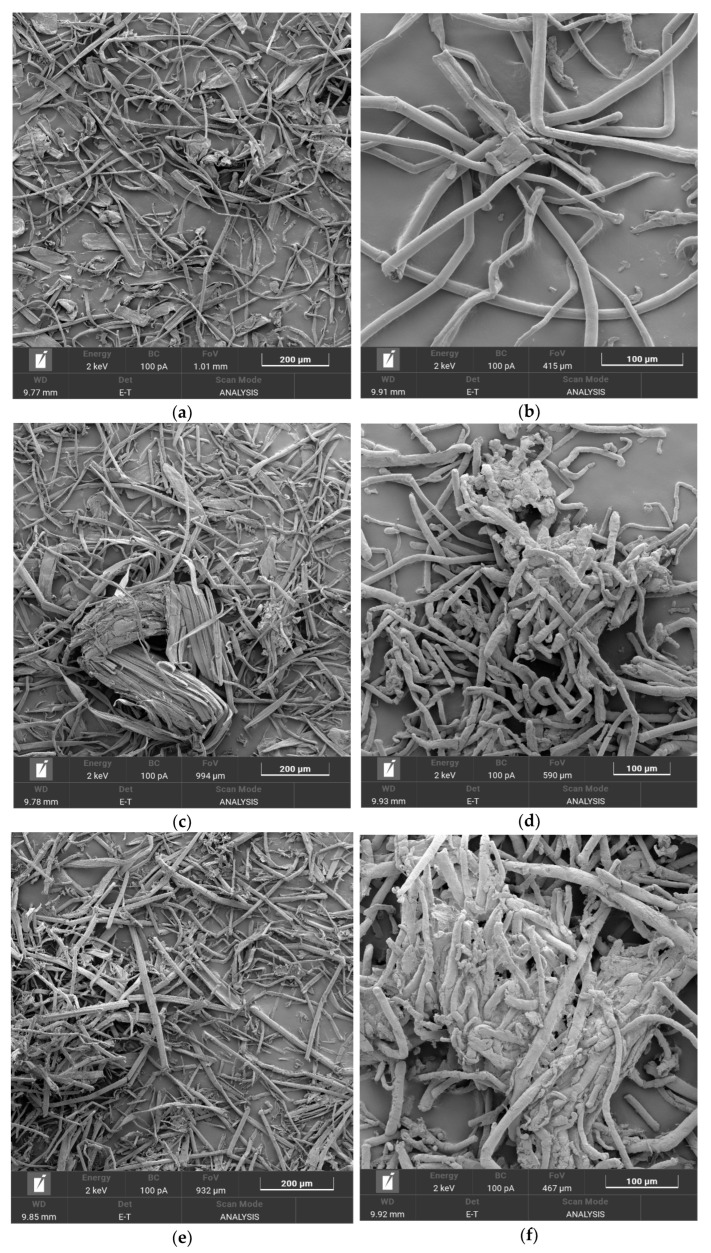
SEM images: (**a**,**b**) initial *Miscanthus* cellulose and CN from it; (**c**,**d**) cellulose after 2 h hydrolysis and CN from it; (**e**,**f**) cellulose after 8 h hydrolysis and CN from it; and (**g**,**h**) cellulose after 24 h hydrolysis and CN from it. Pores on cellulose fibers after hydrolysis are indicated in the SEM images.

**Figure 2 polymers-16-00042-f002:**
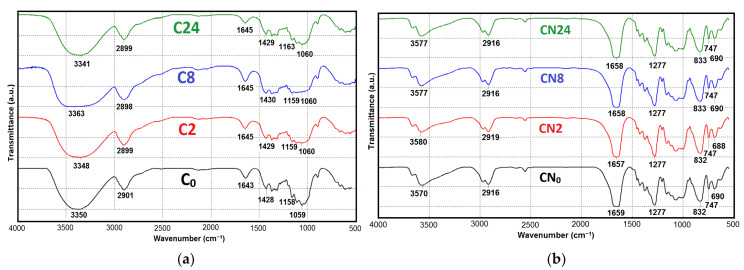
FTIR spectra: (**a**) initial *Miscanthus* cellulose (C_0_) after 2 h hydrolysis (C2), 8 h hydrolysis (C8), and 24 h hydrolysis (C24); and (**b**) CN from original cellulose (CN_0_), after 2 h hydrolysis (CN2), 8 h hydrolysis (CN8), and 24 h hydrolysis (CN24).

**Figure 3 polymers-16-00042-f003:**
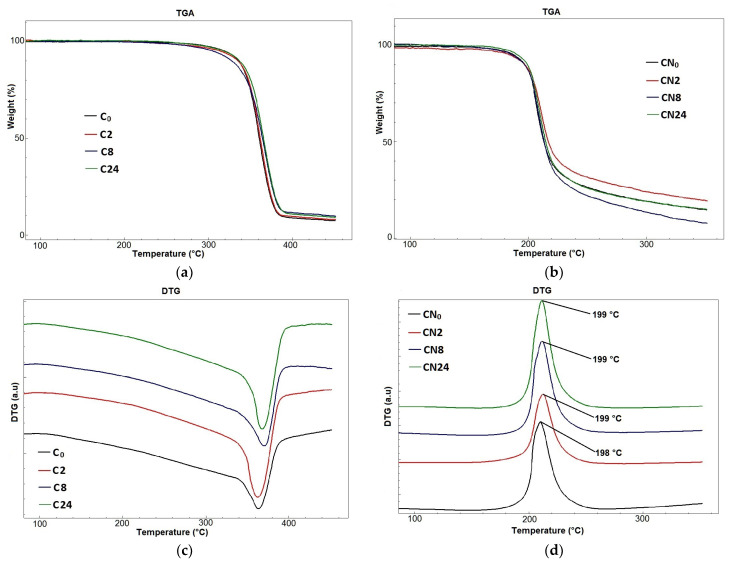
TGA and DTG thermograms of cellulose samples: (**a**,**c**) original cellulose (C_0_), cellulose after 2 h hydrolysis (C2), cellulose after 8 h hydrolysis (C8), cellulose after 24 h hydrolysis (C24); and their cellulose nitrates (**b**,**d**): cellulose nitrate from the original cellulose (CN_0_), cellulose nitrate from cellulose after 2 h hydrolysis (CN2), cellulose nitrate from cellulose after 8 h hydrolysis (CN8), and cellulose nitrate from cellulose after 24 h hydrolysis (CN24).

**Figure 4 polymers-16-00042-f004:**
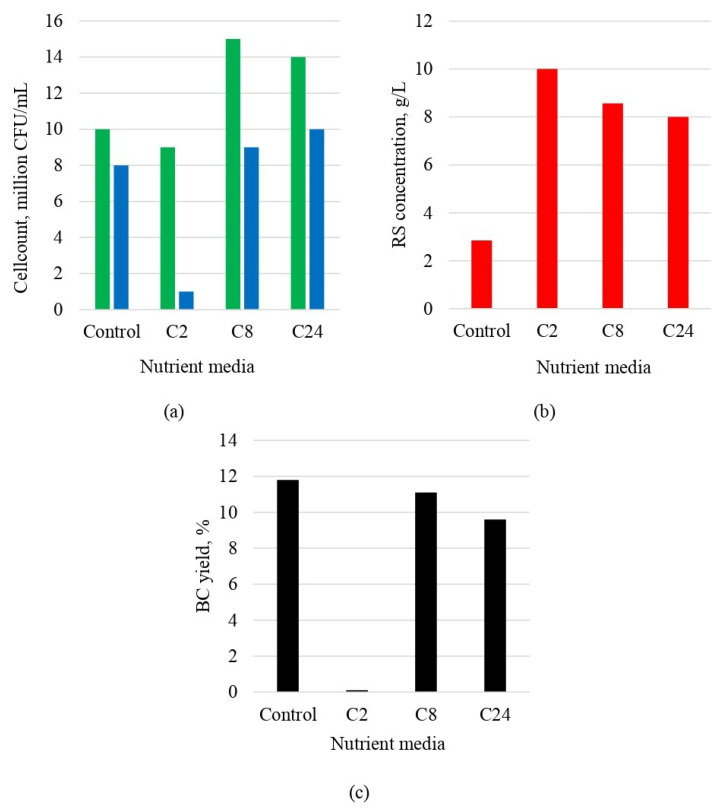
Indicators of BC biosynthesis after 10 days of cultivation in the control and in hydrolyzates C2, C8, and C24: (**a**) 
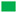
 yeast and 
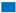
 acetic acid bacteria count in the nutrient medium; (**b**) RS concentration; (**c**) BC yield.

**Figure 5 polymers-16-00042-f005:**
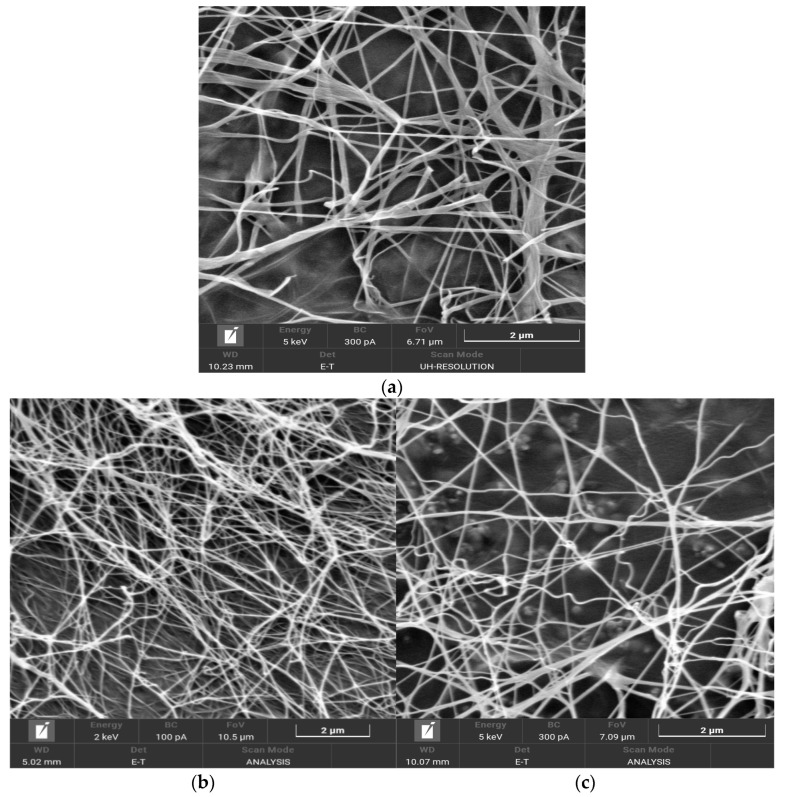
SEM images of BCs derived from the enzymatic hydrolyzates: (**a**) control; (**b**) C8; and (**c**) C24.

**Table 1 polymers-16-00042-t001:** Enzymatic hydrolysis results for *Miscanthus* cellulose (stage 1).

Hydrolysis Time, h	Weight Loss, %	DP	DC, %	RS Content, g/L
0	–	1770 ± 10	64 ± 5	0 ± 0.0
2	32	1490 ± 10	72 ± 5	12.9 ± 0.2
4	47	1620 ± 10	72 ± 5	16.8 ± 0.5
6	48	1640 ± 10	73 ± 5	18.8 ± 0.5
8	50	1640 ± 10	74 ± 5	21.8 ± 0.5
24	58	1750 ± 10	75 ± 5	26.5 ± 0.5
32	65	1780 ± 10	76 ± 5	27.8 ± 0.5
48	66	1790 ± 10	76 ± 5	29.8 ± 0.5

**Table 2 polymers-16-00042-t002:** Enzymatic hydrolysis results for *Miscanthus* cellulose (stage 2) (C—cellulose sample).

Indicator	Hydrolysis Time, h		
	2	8	24
Name	C2	C8	C24
Cellulose weight after hydrolysis, g	3.07 ± 0.0001	3.29 ± 0.0001	3.36 ± 0.0001
DP	1510 ± 10	1670 ± 10	1760 ± 10
DC, %	72 ± 5	74 ± 5	75 ± 5
RS concentration in hydrolyzate, g/L	13.4 ± 0.2	22.8 ± 0.5	27.5 ± 0.5

**Table 3 polymers-16-00042-t003:** Basic functional properties of CN samples.

Sample	Characteristics			Yield *, %
	N, %	Viscosity of 2% Solution in Acetone, mPa·s	Solubulity in Mixed Alcohol/Ester Solvent, %	
CN_0_ (initial)	12.20 ± 0.05	120 ± 1	41 ± 2	142
CN2	11.35 ± 0.05	94 ± 1	79 ± 2	116
CN8	11.63 ± 0.05	113 ± 1	75 ± 2	123
CN24	11.83 ± 0.05	119 ± 1	94 ± 2	131

* Note: The yield calculated after CN was open air-dried.

## Data Availability

Data are contained within the article and Appendix A.

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
