# Peer review of "Simultaneous Production of Cellulose Nitrates and Bacterial Cellulose from Lignocellulose of Energy Crop"

_polymers, 2023, doi:10.3390/polym16010042_

Round 1

Reviewer 1 Report

Comments and Suggestions for Authors

The Ms reports some results on CN and BC preparation that might be considered for publication if the characterization is improved. The novelty is relatively low in the present stage as similar results can be already found in the literature, nevertheless by improving the characterization, the MS can be considered for publication.

-          In line 144 there is an equation and it should be labeled as eq 1. Therefore all the labels of other equations should be changed accordingly.

-          Number of references is huge (100!) that would be fine for a review but in a paper that is not justified.

-          The characterization must be improved. NMR spectra, Mw determination have to be included for the obtained polymer.

-          Nitrate percentage in the cellulose polymer should be determined in a accurate way.

Reviewer 2 Report

Comments and Suggestions for Authors

The introduction needs to be modified to underscore the significance of bacterial cellulose and specify the intended application of this work. Bacterial cellulose offers notable advantages, such as purity, cost-effectiveness, robust mechanical properties, and electrical conductivity. Emphasizing these features is crucial for capturing the reader's interest in the subject matter. I recommend citing this article which provides evidence of the exceptional mechanical strength and electrical conductivity of pure bacterial cellulose, thereby strengthening the relevance and showcasing the potential applications of this material. (https://www.mdpi.com/2073-4360/15/3/643)

Reviewer 3 Report

Comments and Suggestions for Authors

1.     Why does fiber volume increase after nitration?

2.     Page 5. Table 1, why Miscanthus cellulose wt. Loss increases with hydrolysis time; however, DC% shows a slower declination. Explain it.

3.     Page 11, It is mentioned that, furthermore, as the hydrolysis process progresses, the number of holes on the cellulose fibers increases (Figure 1g). However, the holes aren’t shown or seem to appear on the fiber. Provide an enlarged view of the fibers so one can clearly visualize pore presence. Replace the hole word with pores.

4.     Why C24 shows minimum RS concentration in comparison to C2 (Figure 4 [b])

5.     Why is BC yield the highest in a controlled environment, and C2 shows the least yield?

6.     Figure 2 is wrong. Redraw it. Why are the x and y labels missing?

7.     Figure 5 is a blur. Replace it.

8.     Figure 4: Do not insert Excel file data in the manuscript. Redraw them in proper software like origin. Replace Figure 4.

9.     Page 1, Line 12: what does x mean as mentioned, Miscanthus × giganteus.

10.  Page 1, Line 12: Rewrite statement 1. Herein, we explored whether two standalone products .....it is not correct.

11.  Page 2, Line 66. Rewrite statement. It is confusing. Miscanthus sacchariflorus (Maxim.) var. Soranovskii

12.  Page 2, Lines 73-75. This line is quoted representing 8 References. Include only the most significant reference representing this statement.

·       "while the others report extensive  information including basic functional, supramolecular, morphological and energetic characteristics of the biopolymers synthesized from a specific feedstock type [9,17-19,21,  22,28, 29].

Similarly, on Page 2, Line 80, this single statement is taken from 10 references. Correct it, and don’t insert and quote unnecessary references.

·       with hydrolyzates from pretreated lignocellulose being examined as the nutrient medium [39-49]

·       as well  as their wide applications ranging from paper industry to biosynthesis products [60-67]

Remove this kind of irrelevant reference quoting throughout the manuscript.

13.  Page 3, Line 111. Use the entire word before using its abbreviation., TAPPI.

14.  Page 5, Line 203. Replace the cooking word with boiling.

15.  Page 17, Line 585: Add space and write in a consistent manner throughout the manuscript. Miscanthus ×giganteus cellulose

16.  Page 17, Line 614: Grammatical mistake, correct it. ....CNs and BC is has been applied fo

17.  Page 7, lines 314-318. Delete this paragraph. It is irrelevant in the results section.

·       Authors should discuss the results and how they can be interpreted from the perspective of previous studies and of the working hypotheses. The findings and their implications should be discussed in the broadest context possible. Future research directions may also be highlighted.

18.  Writing style is very difficult.

19.  Self-citation is observed; 16 References were quoted from the author's previous work; I will not recommend it till it needs some important worth statement to quote; otherwise, delete them [ Ref 24, Ref 25, Ref 26, Ref 27, Ref 28, Ref 29, Ref 33, Ref 47, Ref 48, Ref 49, Ref 52, Ref 66, Ref 67, Ref 87, Ref 88 and Ref 94].

20.  24% Plagiarism is observed. Remove it to <10 %. It is a serious issue in research articles.

Comments on the Quality of English Language

Page 7, line 332. Those authors explain it by the fact .... It’s wrong. Rewrite the statement.

Page 9. line 385. Based on the obtained data (Table 3). Rewrite it, i.e., As mentioned in Table 3.

Preposition and grammatical mistakes at different places in the manuscript.

Round 2

Reviewer 2 Report

Comments and Suggestions for Authors

All comments have been addressed. I accept publication to this journal. Good luck.

Reviewer 3 Report

Comments and Suggestions for Authors

 Accept in present form.